# Shifts of Leaf Litter-Induced Plant-Soil Feedback from Negative to Positive Driven by Ectomycorrhizal Symbiosis between *Quercus ilex* and *Pisolithus arrhizus*

**DOI:** 10.3390/microorganisms11061394

**Published:** 2023-05-25

**Authors:** Maurizio Zotti, Giuliano Bonanomi, Luigi Saulino, Emilia Allevato, Antonio Saracino, Stefano Mazzoleni, Mohamed Idbella

**Affiliations:** 1Department of Agricultural Sciences, University of Naples Federico II, Via Università, 100, 80055 Portici, Italy; giuliano.bonanomi@unina.it (G.B.); luigi.saulino@unina.it (L.S.); eallevat@unina.it (E.A.); antonio.saracino@unina.it (A.S.); stefano.mazzoleni@unina.it (S.M.); mohamed.idbella@unina.it (M.I.); 2Task Force on Microbiome Studies, University of Naples Federico II, 80138 Naples, Italy

**Keywords:** plant-soil feedback, ectomycorrhizas, self-DNA, autotoxicity, leaf litter, root system

## Abstract

Ectomycorrhizas (ECM) are a common symbiotic association between fungi and various plant species in forest ecosystems, affecting community assemblages at the landscape level. ECMs benefit host plants by increasing the surface area for nutrient uptake, defending against pathogens, and decomposing organic matter in the soil. ECM-symbiotic seedlings are also known to perform better in conspecific soils than other species unable to carry the symbiosis, in a process referred to as plant-soil feedback (PSF). In this study, we tested the effects of different leaf litter amendments on ECM and non-ECM seedlings of *Quercus ilex* inoculated with *Pisolithus arrhizus* and how they altered the litter-induced PSF. Our experiment showed that the ECM symbiont induced a shift from negative PSF to positive PSF in *Q. ilex* seedlings by analysing plant and root growth parameters. However, non-ECM seedlings performed better than ECM seedlings in a no-litter condition, indicating an autotoxic effect when litter is present without ECM symbionts. Conversely, ECM seedlings with litter performed better at different decomposition stages, suggesting a possible role of the symbiosis of *P. arrhizus* and *Q. ilex* in recycling autotoxic compounds released from conspecific litter, transforming them into nutrients that are transferred to the plant host.

## 1. Introduction

Ectomycorrhizal (ECM) symbiosis is one of the most extensive symbiotic relationships in terrestrial ecosystems [1]. The ECM relationship is formed by basidiomycetes, fungi, ascomycetes, and mucoromycetes. Plant species that form ECM symbioses usually belong to the Fagaceae, Betulaceae, Pinaceae, and Salicaceae families in temperate climates [2], while ECM symbioses are less common in tropical environments and are formed with plant species of Dipterocarpaceae and Caesalpinioideae in well-drained soils [2,3]. For plants, the advantages of ECM symbiosis involve improved nutrient acquisition from the soil and increased defences against soil-borne pathogens [4,5]. Consequently, such symbiosis modifies plant-plant competitive interactions, favouring those forming ECMs. At the landscape level, it is more common to find monodominant plant communities associated with ECM symbiosis [6]. The monospecific composition of plant communities is also described as an effect of positive plant-soil feedback (PF) [7]. In the case of PF, a plant species conditions its soil to increase the probability of successful conspecific recruitment and competitively exclude heterospecific recruitment. The opposite condition is described as plant-soil negative feedback (NF), in which conspecific recruitment is disfavoured while heterospecific recruitment is more likely to take place. Consequently, the PF favours the establishment of monodominated communities, while the NF triggers the formation of more diverse plant communities [8]. Vegetation patterns formed by NF are more common than those formed by PF [9,10]. For example, at the individual level, rings of clonal plants migrate outward from the negatively conditioned centre [11,12]. In grassland species, plant turnover is driven by NF [13,14]. Furthermore, in the successional process of vegetation, NF conditions in the soil may favour the shift from one plant community to another [15,16,17].

Several pieces of evidence supported different hypotheses regarding the biological mechanisms leading to a plant-soil feedback response. Plant pathogen accumulation beneath parent plants [18], depletion of nutrients [19,20], and the autotoxic effect caused by negative products of organic matter decomposition [21,22,23]. In the case of ECM forests, a non-fingo hypothesis suggests that the symbiosis between the plant and the fungus acts in the way in which one or all the NF mechanisms are transmuted in PF. The probability that a seedling will be inoculated by a suitable ectomycorrhizal fungus is higher near a conspecific tree. Accordingly, an evident advantage is provided to ECM seedlings to avoid the consequences of NF [7,24]. ECMs were documented to increase plant defences against pathogens by establishing a selective mycorrhizosphere. Additionally, the mycelium of EMC fungi extends the nutrient adsorption surface for the plant host, so that the plant benefits from the degradation activity of soil organic and inorganic matter by the fungus [25,26,27,28,29]. Moreover, seedlings growing in the proximity of mother plants have a higher probability of being integrated into the common mycorrhizal network and receiving an advantage through resource trading between adult trees [6,30]. Conversely, few studies have focused on the ability of ECM symbiosis to eliminate the autotoxic effect derived from the decomposition of conspecific organic matter. Several studies showed that seedling growth and survival were reduced in soils with high inputs of leaf litter [31,32,33]. However, during leaf litter decomposition, the effect becomes species-specific, with impaired growth of conspecific seedlings and a shift in effect from phytotoxicity to autotoxicity [23,34]. Litter decomposition patterns and associated autotoxicity offer new insights into plant-soil interaction studies and have the potential to explain plant community patterns at inter-ecosystemic levels, such as grasslands, forests [17,34,35,36], and stream ecosystems [37]. A wide and detailed assessment of the effects of autotoxicity was proposed by Mazzoleni et al. in 2015 [22], indicating that the process is triggered by the release of autotoxic self-DNA. Among the plants tested in their work, they described a self-inhibitory effect also for plants capable of forming ECM symbioses, such as *Quercus ilex* and *Quercus pubescens*, with a NF effect observed in ECM plants despite their monodominant distribution, indicating a PF effect in natural environments. Thus, the theory of leaf litter autotoxicity needs to be verified, including the presence of symbioses that may influence the natural distribution of plants. Evidence suggests that ECM symbiosis promotes the exploitation of leaf litter for the plant through the saprobic activity of the fungus [38,39,40], hypothetically shifting the effect described by Mazzoleni et al. [22,41] into a nutritive advantage for the plant bearing the symbiosis. However, knowledge on the ability of ECM symbiosis to reduce or eliminate the detrimental effects of litter is lacking in the literature. Moreover, most of the works studied this effect as plant-soil interaction rather than symbiotic system-soil, which is a consistent bias considering that plants should be studied as holobionts [42].

From this point of view, we consider the different responses in morphometrics and plant growth as important indicators of the relationship between the symbiotic system and environmental conditions in the soil. This is particularly true for the enrichment with conspecific leaf litter at different ages of decomposition. In this context, we hypothesise that the ECM symbiosis can convert NF to PF due to plant litter. To test this hypothesis, we used a symbiotic system composed of *Q. ilex* seedlings and *Pisolithus arrhizus,* considering it a natural and widespread association in Mediterranean regions [43,44,45]. Previous studies demonstrated that conspecific litter inhibits *Q. ilex* seed germination and root growth in the early stage [41,46], but no studies have investigated the litter impact on seedling growth and root structure. The specific aims of the study were:Assess the effect of litter-induced plant-soil feedback across different decomposition stages in seedlings with and without ECM symbiosis of *P. arrhizus*.Connect the effect of litter-induced plant-soil feedback response with changes in the root system growth and structure upon addition of ECM inoculum.

## 2. Materials and Methods

### 2.1. Production of Mycorrhized and Non-Mycorrhized Quercus ilex Seedlings

All materials were collected in the evergreen holm oak urban forest of Parco Gussone (40°48′40.3″ N, 14°20′33.8″ E, 75 m a.s.l.; about 1 km from the coast) in the Bourbonic royal palace of Portici. The site is an old-growth evergreen *Quercus ilex* forest established on the lava field approximately 300 years ago. The overstory tree layer is dominated by *Q. ilex* with an average stand height of 16 m, while the understory is mainly composed of shade-tolerant deciduous *Fraxinus ornus* and evergreen *Laurus nobilis* species [47].

The soil is shallow with andic properties and overlies pyroclastic deposits from the eruptions of Vesuvius in 1631 CE. The climate is Mediterranean, with humid winters and dry summers, total annual rainfall of 929 mm (290, 200, 89, and 348 mm in winter, spring, summer, and fall, respectively), and monthly average temperatures ranging from 11 °C (January) to 26 °C (August) [48].

The fungal material for inoculation was provided by ID Forest laboratories (http://idforest.es/, accessed on 15 June 2018, Spain) and consisted of certified spores of *Pisolithus arrhizus*. Acorns of *Q. ilex* were collected from plants on which the presence of sporophores of *Pisolithus* spp. was observed. Acorns were selected, and those that showed damage, lesions, or structural defects were discarded. Seeds were dried for several days and then stored at a temperature of 4°C for vernalization. Given the nondormant nature of *Q. ilex* acorns, a period of one week was considered sufficient to achieve their simultaneous germination. Prior to germination, seeds were sterilised for 5 min in sodium hypochlorite at a concentration of 1% (*v*/*v*). After sterilisation, seeds destined for inoculation were separated from those not destined to form ECM symbiosis to produce groups of mycorrhized and non-mycorrhized *Q. ilex* seedlings. For each group, acorns were germinated in separate plastic boxes 60 × 40 × 40 cm. A sterilised mixture of peat and vermiculite in a 3:1 ratio was used as the substrate for germination. The germination of the oak acorns occurred above ground and uniformly 5 days after they were placed in plastic boxes. Seeds that had not germinated after ten days were discarded. The inoculation phase consisted of the addition of 5 g of *P. arrhizus* spores to the water used for irrigation. The inoculum was sprayed on the germinated seedlings, and the inoculation was repeated every 4 days for a total of three times. After one month of growth in lab conditions, seedlings were transplanted under experimental conditions. For ECM seedlings, only those showing ectomycorrhizal root tips were included in the experiments. *Q. ilex* seedlings were checked for successful formation of ECM root tips by inspecting them under a dissection microscope at 4× magnification. Seedlings in the range of 15 to 30% ECM root tip colonisation (based on the total number of root tips in the root system) were included in the experiment and compared with non-inoculated seedlings. Seedlings falling below or exceeding the colonisation threshold were not included in the experiment.

### 2.2. Rhizotron Experiment

A rhizotron experiment was conducted to investigate the impact of conspecific litter and the presence of ECM fungi on the development of *Q. ilex* root systems. The rhizotrons were composed of modules measuring 100 cm in depth, 25 cm in width, and 5 cm thick. One side of the rhizotron was covered in transparent polyethylene material, while the other side was covered in a rigid, opaque polycarbonate layer. The narrower side consisted of foam bands (Figure 1). The rhizotron modules were sterilised by washing with 1% sodium hypochlorite and rinsed with water twice. Autoclaved soil was used to fill the rhizotrons, and 2% litter was added to the total soil amount (~2 kg). Fresh and aged conspecific litter (decomposed for 0 and 120 days, respectively) were added to investigate the impact of litter age. Control groups without litter were also prepared. The three litter treatments were applied to both ECM and non-ECM plants (Figure 1). The 120-day decomposed litter was produced under laboratory conditions in 50 × 30 × 30 cm plastic trays that were periodically watered at a temperature of 20°C. To promote decomposition of the leaf litter, a soil sample was collected in correspondence with the leaf litter sampling site. Soil inoculum was added using the slurry method, according to Bonanomi et al. in 2019 [49]. After litter addition, *Q. ilex* seedlings with and without ECM symbiosis were transplanted and covered for three days to avoid hydric stress during transplantation. The rhizotrons were placed at an inclination of approximately 60° to promote roots’ gravitropic growth towards the transparent plastic layer, and roots were traced using different coloured indelible pencils for each sampling date (Figure 1). The experiment lasted for two months (the period required for the roots to reach the base of the mesocosm), after which parameters requiring destructive sampling were collected. A total of 18 rhizotrons were prepared according to the experimental design. Data on root system depth and width, area, total root length, total length of fine roots, and number of fine roots were collected every 15 days. At the end of the experiment, shoot, root, and total biomass were measured by destructive sampling. Additionally, the ratio between root and shoot biomass, depth and width of the root system, root density as the number of fine roots per unit area, the ratio between root width and biomass, and the ratio between DW and biomass were calculated (Appendix A).

### 2.3. Statistical Analysis

The significant changes in plant and root growth parameters of *Q. ilex* seedlings were tested with and without ECM symbiosis using three different litter decomposition regimes (no litter, 0 days, and 120 days). GLM and the specific significant differences between experimental groups were assessed using Duncan’s test at α = 0.05.

To observe the effect of conspecific litter on root system proliferation with or without ECM symbionts, the effect size was calculated by subtracting the value of *Q. ilex* seedlings growing in the presence of conspecific litter from the corresponding value observed in the no litter treatment and dividing by the standard deviation of the data population [50]. Effect size values were calculated for each of the variables used in the experiment. The effect size values were first used to observe the ordination of seedlings according to the effect of litter on root systems with or without ectomycorrhizal symbionts using principal component analysis (PCA). Later, data for ECM and non-ECM seedlings of *Q. ilex* were separately observed through heatplots with variables ordered by hierarchical clustering of Euclidean distance of effect size values for seedlings growing in soil amended with fresh (0 days) and aged (120 days) litter.

The increase rate in total length, total length of fine roots, and number of fine roots were analysed to examine the effects of the presence of symbionts, the presence and absence and duration of decomposition of self-litter added, and the timing of sampling. Other parameters explaining root proliferation and the trajectory assumed across the dates of sampling were analysed using principal component analysis (PCA). All statistical analyses and plots were performed using Primer 7 and Statistica 10 software.

## 3. Results

Digital imaging of *Q. ilex* seedlings showed generalised changes in root structure among treatments and the presence/absence of ECM symbiosis (Figure 2). In detail, seedlings growing with ECM symbionts had higher root growth compared to non-ECM seedlings when growing with conspecific litter compared to the corresponding treatment of non-ECM seedlings. Among these, ECM seedlings growing with fresh litter (0 days) showed the highest root biomass, followed by seedlings growing in decomposed litter (120 days). For non-ECM seedlings, lower root biomass was recorded for those growing in decomposed litter (120 days), with similar values at 0 days of decomposition. In the absence of litter, higher root biomass was produced in non-ECM seedlings (Figure 3a).

The treatments used in the experiment showed a similar effect on total root length, the number of fine roots, and the total length of fine roots. The highest values of total root length, number of fine roots, and total length of fine roots were observed for ECM seedlings growing in the conspecific litter, both fresh and decomposed. Meanwhile, non-ECM seedlings growing in conspecific litter showed a lower level of total root length, the number of fine roots, and the total length of fine roots. Among seedlings growing without litter, higher total root length, the number of fine roots, and the total length of fine roots were observed for non-ECM seedlings when compared to the ECM (Figure 3b–d). ECM plants developed a larger area of the root system compared to non-ECM seedlings. The only exception is made for non-ECM seedlings that develop similar area values to the ECM ones (Figure 3e).

The density of fine roots was similar between ECM seedlings; meanwhile, variation was observed for the non-ECM plants. The lowest level of fine root density was observed in the non-ECM seedlings growing in the decomposed litter, while the highest level was recorded in the same group of seedlings growing in the absence of litter (Figure 3f).

PCA of the effect sizes measured as the difference in root development parameters from the seedlings growing in fresh and decomposed litter against values of those in the absence of litter showed an inverse response to litter supplement between ECM and non-ECM seedlings (Figure 4).

Commonly, ECM plants growing in litter developed higher values for density of the root system, total length and number of fine roots, total root length, area of the root system, and the ratio between root length and area compared to plants without litter. Inversely, the DW/RB ratio was lower compared to plants growing without litter. Specifically, for how the decomposition stage of litter affects the development of the root system in ECM seedlings, we observed that ECM seedlings in fresh litter developed root systems with a lower width/root biomass ratio. Inversely, seedlings in decomposed litter developed a higher width of the root system and R/S and a lower depth of the root system and D/W ratio (Appendix A).

For non-ECM plants, we observed that seedlings growing in conspecific litter developed a lower depth of the root system, the ratio between root length and area, the density of fine roots, the number of fine roots, the root biomass, the DW/RB ratio, the total length of fine roots, the DW ratio, and the total root length. Decomposition stages of litter produced a higher area and width of the root system and a higher width/biomass of the root system ratio in seedlings growing with decomposed litter. Inversely, seedlings growing in fresh litter developed a higher ratio of root biomass, root depth, and R/S ratios.

Moreover, nine parameters were measured to obtain information on the shape of the root system and its strategy of proliferation in the soil over time. The data was reduced using Principal component analysis (PCA), explaining around 99.6% of the total variance in the data (PCI: 61.3% and PCII: 29.3%). The multivariate analysis showed that ECM plants are generally more correlated with the development of a wider root apparatus and a larger area of the root system. The same strategy is assumed by litter-enriched non-ECM plants that develop a wider root apparatus but have a lower correlation because of the scant growth. A complete inverse strategy is developed in non-ECM plants growing without litter, which have a higher depth-width ratio, a higher density of fine roots, and a deeper root system (Figure 5).

## 4. Discussion

In the present experiment, we demonstrated the positive relationship between *Q. ilex* seedlings and *P. arrhizus*, one of its elective fungal symbionts. We observed that seedlings growing without the ECM symbiont showed lower growth rates and root proliferation in the litter compared to seedlings growing without the symbiont and without conspecific litter. Differently, when *P. arrhizus* was present, seedlings of *Q. ilex* benefited from the presence of conspecific litter. Overall, the results support our hypothesis that the presence of the ECM symbiont promotes a shift in NF produced by conspecific litter in PF. The findings are consistent with previous research highlighting the positive impact of ECM symbiosis on plant establishment in soil conditioned by conspecifics [7,51].

Regarding the effects of ECM symbiosis on the structure of the root system of *Q. ilex*, our analysis revealed that the roots of ECM seedlings proliferated more widely and extensively in the soil compared to non-ECM seedlings. The difference in root system structure was particularly noticeable when observing ECM plants growing with conspecific litter and non-ECM plants growing without litter, which prefer to grow in a deeper soil layer without the lateral ramification of lateral leading roots that over-expand in a bifid herringbone structure. This root structural change may have arisen due to the proliferation of ECM plants in environments where disturbance or changing factors in root structure are minimised. In ECM plants, symbiosis may have acted as an enhancer of root ramification [52,53]. As a result, the higher number of root tips in ECM plants triggered a cascade mechanism where intraindividual competition between root tips led to a wider root system due to a self-repulsive arrangement [54]. ECM plants likely have an advantage in resource acquisition due to their larger adsorption surface, but only in the presence of conspecific litter.

In estimating the effects of litter decomposition, we observed few changes in ECM plants growing in fresh or decomposed litter, suggesting that litter may have a positive effect on the symbiotic system between *Q. ilex* and *P. arrhizus* regardless of the decomposition stage. Our observation is not consistent with previous findings on the natural feeding strategy of ectomycorrhizal fungi, which preferentially scavenge resources in the soil layer below fresh litter on more decomposed materials [28]. In contrast, fresh litter is dominated by saprotrophic fungi [55]. It could be that the lack of a saprobic community in our experimental system allowed ECM plants to also benefit from fresh litter in the soil. Another explanation for this phenomenon could be based on a specific ability of *P. arrhizus,* as most ECM fungi still have obscure mechanisms of action [56,57]. Notably, ECM seedlings growing with conspecific litter developed deeper root systems compared to non-ECM plants. The presence of litter with an autotoxic effect can impair the exploration of soil horizons with higher water availability, exposing plants to possible hydric stresses. This effect should be investigated in a natural environment, as in our experimental system, water availability does not constitute a limiting factor.

In contrast, we observed variable trends in root proliferation over time in non-ECM plants when amended with fresh or decomposed litter. Although the structure and biomass of the root system at the end of the experiment were similar between seedlings in decomposed and fresh litter, the former had a higher proliferation of the root system during the first 15 days of growth, followed by a lower ability to form new root parts on the following sampling date. Conversely, fresh litter exhibited lower proliferation of the root system in the first 15 days, followed by a net increase in the ability to produce new root parts on subsequent sampling dates. Generally, these trends indicate that the root system loses the ability to proliferate in some portions of the roots during the experiment, suggesting that the factor causing this effect is persistent over time. Conversely, in fresh litter, the effect on root system proliferation is ephemeral during the first 15 days. This finding is consistent with the results of Mazzoleni et al. [22,41], who suggested a negative species-specific effect of decomposed litter on the growth of conspecific seedlings. Interestingly, the long-lasting negative effect of decomposed litter on seedling growth suggests the presence of a stable factor affecting root proliferation, which could be autotoxic self-DNA. Further research, including testing a higher number of plants and the addition of heterospecific litter, which is lacking in the present experiment, is necessary to clarify this effect.

Finally, we can draw some conclusions from our experiment about the mechanisms by which ECM symbiosis shifts an NF to a PF generated by conspecific litter. We suspect that litter may become a nutrient resource for the fungus involved in the symbiosis, as has been observed in the symbiosis between *Paxillus involutus* and *Betula pendula* [39,40], for the ECM fungus *Cortinarius glaucopus* [58], and at the ectomycorrhizal community level [28]. Additionally, the presence of the fungal symbiont, which envelops the fine root tips with a dense hyphal mantle, may protect against a potential autotoxic effect of leaf litter [59,60]. We mainly support the nutrient source hypothesis, where the fungus can use the nutrient immobilised in leaf litter to degrade potential autotoxic compounds. The last is supported by the assumption that we observed increased root proliferation in ECM seedlings with conspecific litter. Conversely, if the fungal symbiont acts only as a protective layer surrounding root tips, it should not affect plant growth compared to ECM seedlings growing without litter. The presence of the basidiomycetous symbiont may prevent the decomposition of conspecific litter by releasing its complex enzymatic arsenal [61,62]. Furthermore, the oxidative ability of basidiomycetous fungi from Fenton chemistry can contribute to the degradation and destabilisation of a wide range of molecules in organic matter [28,63], including long-lasting autotoxic compounds that act at a species-specific level such as self-DNA. In agreement with the literature, the ECM fungal community is more abundant in the intermediate layer of the forest floor, where there is a concomitant increase in recalcitrant compounds, but also in self-DNA. Consequently, it could be suggested that the ability of the plant to support the symbiosis is the result of an evolutionary adaptation to NF. The function of an associated ECM community shifts from NF to PF, allowing the formation of monodominant stands of the plant species in question.

In addition, it would be interesting to verify in future studies whether the effect observed in the present work is maintained when a different combination of ECM symbiosis and amendment of conspecific and heterospecific litter is used, as well as whether the effect is the same in ECM plant species that form monodominant or mixed stands. Furthermore, it would be worthwhile to investigate whether the addition of conspecific litter can induce stress in plants and thus promote the formation of ECM symbiosis.

## 5. Conclusions

In our experiment, we observed that the presence of the ECM symbiont *P. arrhizus* in *Q. ilex* resulted in a shift in NF to PF from non-ECM to ECM fungi generated by conspecific litter. We speculate that this effect is likely due to the increased ability of ECM plants to scavenge resources in the soil due to their broader and more extensive root systems, but also to the capability of their root systems to explore deeper root layers when the effect of litter is minimised by ECM symbiosis or is absent. We also observed that non-ECM seedlings had reduced growth and root proliferation compared to non-ECM seedlings growing without litter, indicating an autotoxic effect of conspecific litter. In these plants, the addition of litter produces similar results but with a different temporal pattern depending on the degree of decomposition of the litter material. It could be suggested that the ECM symbiosis between *P. arrhizus* and *Q. ilex* plays a role in recycling autotoxic compounds released from conspecific litter by converting them into nutrients that are passed on to the plant partner.

## Figures and Tables

**Figure 1 microorganisms-11-01394-f001:**
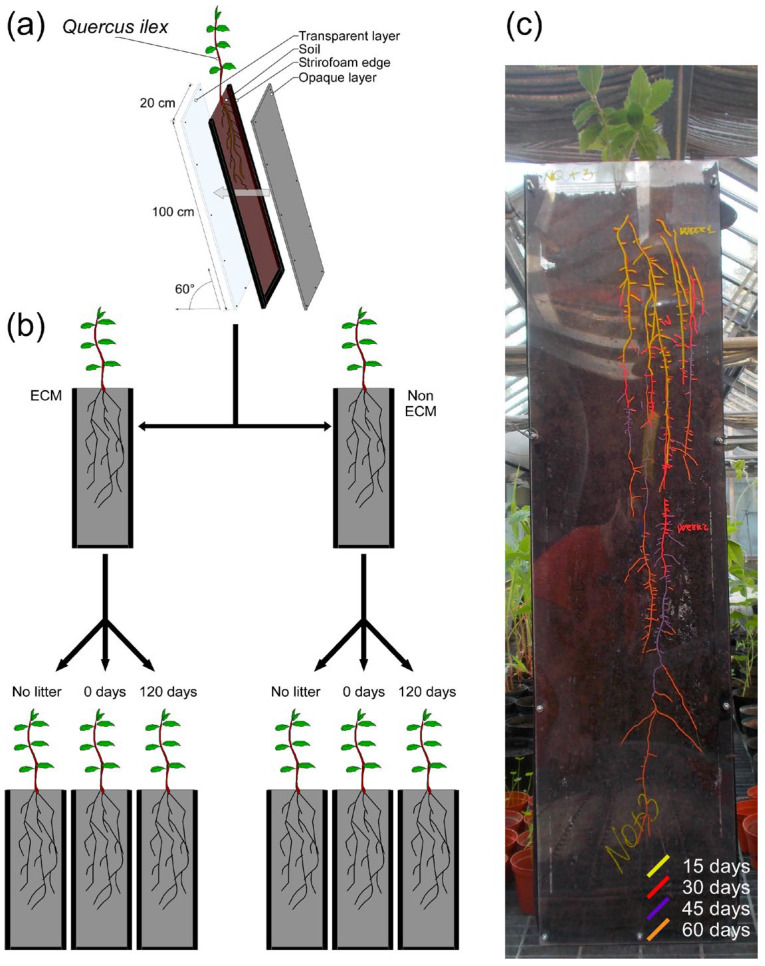
Representation of the rhizotron experiment. (**a**) Module structure; (**b**) experimental design and combination of soil litter amendments used in the experiment on *Q. ilex* seedlings with or without ectomycorrhizae of *P. arrhizus.* (**c**) In this example of the root system of a *Q. ilex* seedling growing in the rhizotron, roots were marked with differently coloured indelible pencils. Different colours represent the root visible after 15 days (yellow), 30 days (red), 45 days (violet), and 60 days (orange).

**Figure 2 microorganisms-11-01394-f002:**
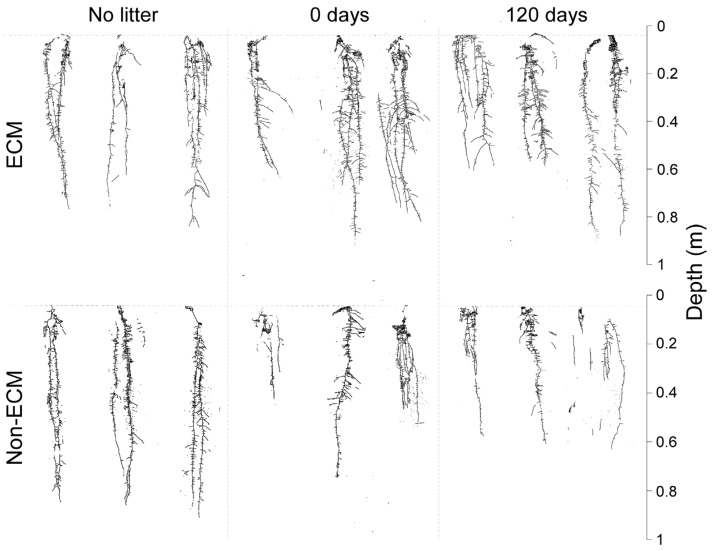
Scan of root systems of *Q. ilex* seedlings growing in the presence (**up**) and absence (**down**) of the ectomycorrhizal symbiosis of *P. arrhizus* with three different types of litter regimes: fresh litter (0 days of decomposition), aged litter (120 days of decomposition), and without litter.

**Figure 3 microorganisms-11-01394-f003:**
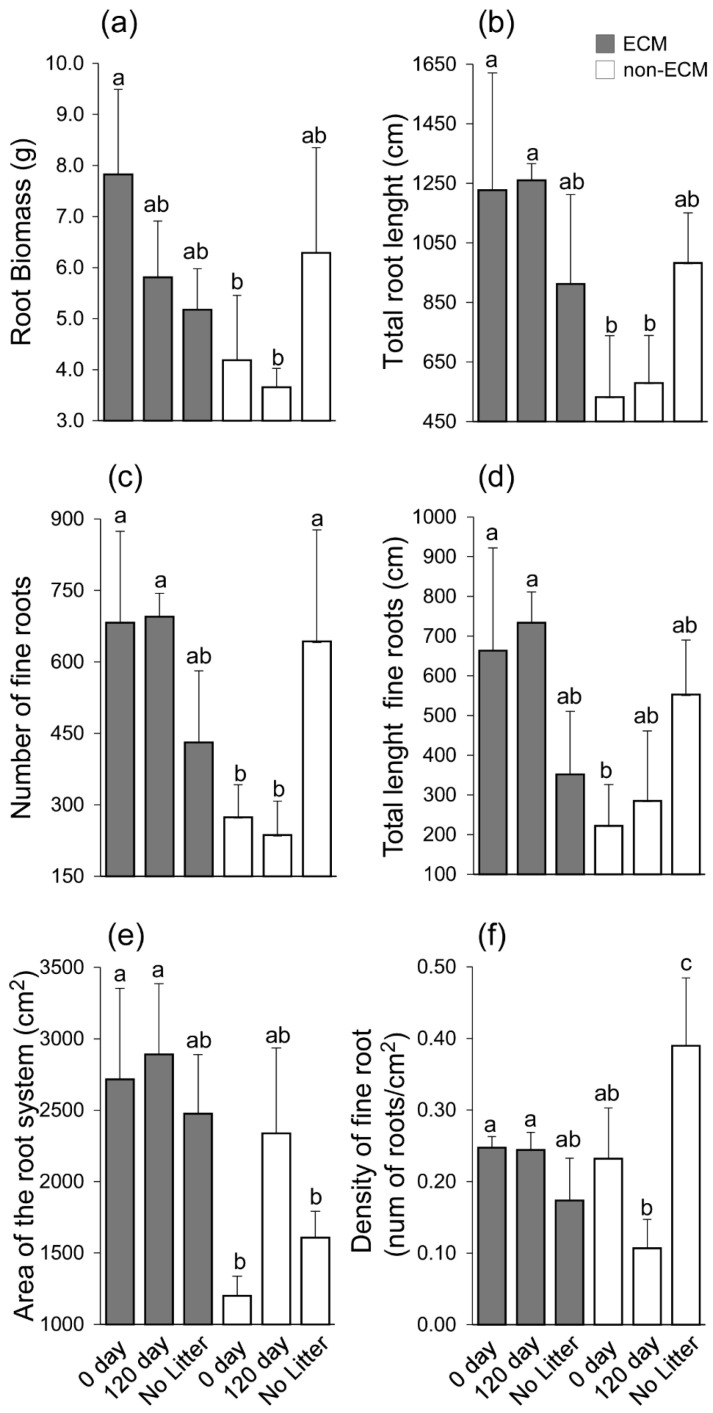
Barplot showing root biomass (**a**), total root length (**b**), number of fine roots (**c**), total length of fine roots (**d**), area of the root system (**e**), and density of fine roots (roots/cm^2^) (**f**). Lecter indicates significant differences between each experimental unit. Significance was assigned according to the Duncan post-hoc test for *p*-values < 0.05. A statistical test was conducted on 18 *Q. ilex* seedlings with three replicates for each treatment. Letters on each bar indicate significant statistical differences between treatment.

**Figure 4 microorganisms-11-01394-f004:**
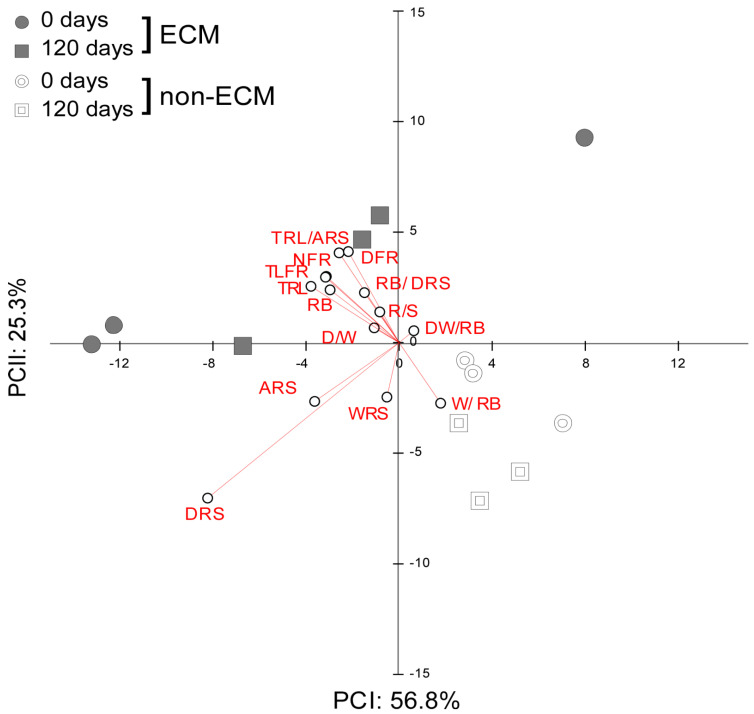
PCA of effect size from *Q. ilex* seedling biometric variables of the root system. Each of the variables was rescaled according to its effect on litter treatment in both ECM and non-ECM conditions for soils enriched with fresh (0 d) and aged (120 d) conspecific litters. Variables abbreviations refer to: width of the root system (cm) WRS; depth of the root system (cm) DRS; area of the root system (cm^2^) ARS; total length of fine roots (cm) TLFR; number of fine roots NFR; total root length (cm) TRL; root biomass (g) RB; root length/area TLFR/ARS; root biomass/root depth RB/DRS; root width/root biomass WRS/RB; density N° root/area NFR/ARS; root biomass/shoot biomass R/S; depth/width of the root system D/W; depth/width/root biomass DW/RB.

**Figure 5 microorganisms-11-01394-f005:**
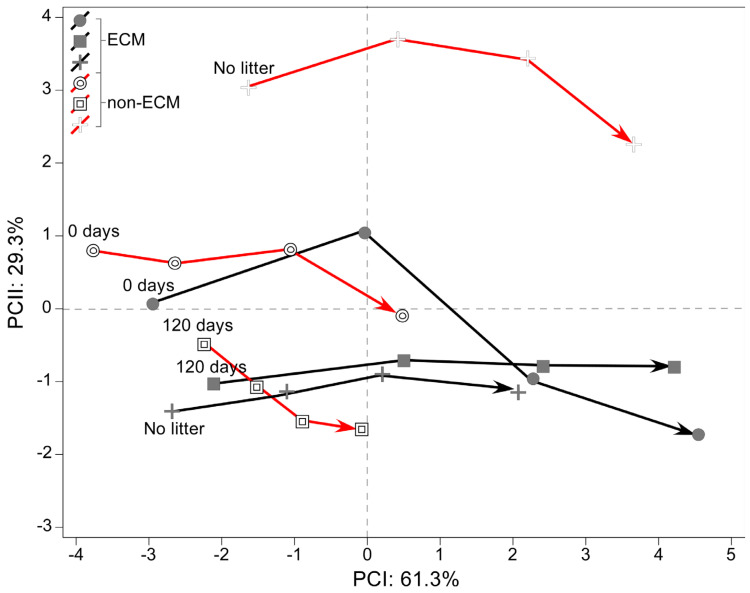
Principal component analysis (PCA) explains variation in root system associated with ectomycorrhizal and non-ectomycorrhizal *Q. ilex* seedlings growing with different litter regimes. In the plot, the trajectories of associated loadings are shown for each sampling date.

## Data Availability

The data presented in this study are available upon request from the corresponding author.

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
