# Peer review of "Shifts of Leaf Litter-Induced Plant-Soil Feedback from Negative to Positive Driven by Ectomycorrhizal Symbiosis between Quercus ilex and Pisolithus arrhizus"

_microorganisms, 2023, doi:10.3390/microorganisms11061394_

Round 1
Reviewer 1 Report
Quercus ilex ectomycorrhizal symbiosis with Pisolithus arrhizus shifts plant-soil feedback from negative to positive.
In this study, researchers investigated the impact of leaf litter on the growth of Quercus ilex seedlings in the presence and absence of the beneficial fungus Pisolithus arrhizus. The study aimed to answer if/how the ECM alters litter-induced plant-soil feedback (PSF). Overall, the researchers found that the presence of ECM symbionts promoted a shift from NF produced by conspecific litter into PSF. This study highlights the benefits of ECM and the need for more research on plant-fungal interactions. However, there are a few issues that the authors need to address:
o Is Q. ilex a natural host of Pisolithus arrhizus? If yes, that needs to be explicitly mentioned and cited. If not, what is the benefit of understanding this non-natural relationship? How do you prevent possible invasiveness from these fungal species if you try to replicate this in the field?
o In-text reference or discussion of Figure 2 seems missing. Please consider merging them with either Figure 1 or 3 and/or mention them in the results and discussion.
o The authors should include the number of biological replicates used for the statistical test in the Figure 3 legend.
o The statement in line 77-78 need to be cited.
o There are several typos that need to be corrected, including "bioassay?" in line 93, a double period in line 107 (" . ."), a missing "Q" in line 121 ("Q. ilex"), and an unintentional period after "non-mycorrhized" in lines 125-126. Additionally, there are several instances of double spaces (e.g., line 144, line 149, line 290) and some with no space (such as line 157).
Author Response
In this study, researchers investigated the impact of leaf litter on the growth of Quercus ilex seedlings in the presence and absence of the beneficial fungus Pisolithus arrhizus. The study aimed to answer if/how the ECM alters litter-induced plant-soil feedback (PSF). Overall, the researchers found that the presence of ECM symbionts promoted a shift from NF produced by conspecific litter into PSF. This study highlights the benefits of ECM and the need for more research on plant-fungal interactions. However, there are a few issues that the authors need to address.
We would like to thank the reviewer for their comments and thoughtful feedback, which we have treated with utmost diligence and attention.
- Is Q. ilex a natural host of Pisolithus arrhizus? If yes, that needs to be explicitly mentioned and cited. If not, what is the benefit of understanding this non-natural relationship? How do you prevent possible invasiveness from these fungal species if you try to replicate this in the field?
We thank the referee for underlining the missing information in the introduction. The answer is yes, Q. ilex a natural host of Pisolithus arrhizus. The text was then adjusted as follows: “To test this hypothesis, we used a symbiotic system composed of Q. ilex seedlings and Pisolithus arrhizus considering it as a natural and widespread association in Mediterranean regions [43-45].” Lines 93-95. We also amended the text with 3 new references specifying the point.
- In-text reference or discussion of Figure 2 seems missing. Please consider merging them with either Figure 1 or 3 and/or mention them in the results and discussion.
Done, sentence added from line 197 to 198 of result section. “Digital imaging of Q. ilex seedling showed a generalized changes in root structure among treatment and presence/absence of ECM symbiosis (Figure 2). In detail, seedlings…..”
- The authors should include the number of biological replicates used for the statistical test in the Figure 3 legend.
Done.
4.The statement in line 77-78 need to be cited.
To avoid being drafted away from the main work that included the plant species in question i.e., Q. ilex and Q. pubescens, we have omitted the portion that references additional works and the sentence was then modified as follows: “Among the plants tested in their work, they described a self-inhibitory effect also for plants capable of forming ECM symbioses, such as Quercus ilex and Quercus pubescens; with a NF effect observed in ECM plants despite their monodominant distribution, indicating a PF effect in natural environments..” Lines 76-79.
- There are several typos that need to be corrected, including "bioassay?" in line 93, a double period in line 107 (" . ."), a missing "Q" in line 121 ("Q. ilex"), and an unintentional period after "non-mycorrhized" in lines 125-126. Additionally, there are several instances of double spaces (e.g., line 144, line 149, line 290) and some with no space (such as line 157).
Done. all typos mistakes had been corrected, for details please check the track-version of the manuscript.
Reviewer 2 Report
I believe this manuscript has done well in terms of experimentation, analysis, and writing. The only shortcoming is that the description of the paper's objective is not clear enough. Specifically, there are a few points:
- The title does not mention "leaf litter" and does not indicate what kind of "Plant-soil feedback" is being referred to. Does it refer to the response of root growth to leaf litter decomposition?
- The last two aims in the Introduction do not summarize the paper's objective. It still does not mention what kind of "Plant-soil feedback" is being studied in the paper. It is recommended to make it consistent with the title.
- The description of the effect of soil depth in the article should be strengthened.
- Please describe the specific content of the nutrient source hypothesis or other hypotheses clearly. I seem to have mentioned some content related to allelopathy. Are they part of the "nutrient source hypothesis"?
- After addressing the above issues, please rewrite the conclusion.
Author Response
Referee #2
I believe this manuscript has done well in terms of experimentation, analysis, and writing. The only shortcoming is that the description of the paper's objective is not clear enough. Specifically, there are a few points:
Thank you for your insightful feedback on our manuscript. We appreciate your positive assessment of the experimentation, analysis, and writing.
- The title does not mention "leaf litter" and does not indicate what kind of "Plant-soil feedback" is being referred to. Does it refer to the response of root growth to leaf litter decomposition?
Done. We changed the title to “Quercus ilex ectomycorrhizal symbiosis with Pisolithus arrhizus shifts leaf-litter induced plant-soil feedback from negative to positive.”.
- The last two aims in the Introduction do not summarize the paper's objective. It still does not mention what kind of "Plant-soil feedback" is being studied in the paper. It is recommended to make it consistent with the title.
We rewrite the aims statement in introduction section to clarify the objectives of the work. We thank the referee for the suggestion aimed to increase the understandability of the work. The manuscript was changed as follows: “
- Assess the effect of litter induced plant-soil feedback across different decomposition stages in seedling with and without ECM symbiosis of P. arrhizus; ii. Connect the effect of litter induced plant soil feedback response with changes in the root system growth and structure upon addition of ECM inoculum.”
- The description of the effect of soil depth in the article should be strengthened.
Done. We amended the Discussion section with a paragraph on the differences in soil depth that are mainly affected by the presence of litter (see statistics in Table S1), specific changes were induced as follows: “
Notably, ECM seedlings growing in the presence of conspecific litter developed a deeper root system compared to non-ECM plants. The presence of litter with autotoxic effects may interfere with the exploration of soil horizons with higher water availability, exposing plants to possible water stress. This effect should be studied in natural environments, as water availability is not a limiting factor in our experimental system.
.” Lines 326-331.
- Please describe the specific content of the nutrient source hypothesis or other hypotheses clearly. I seem to have mentioned some content related to allelopathy. Are they part of the "nutrient source hypothesis"?
We thank the referee for underlining the missing part in our discussion. We clearly state that nutrient source hypothesis involves the ability of the fungus to scavenge nutrients immobilized in leaf litter and eventually degrade autotoxic compounds. Specific changes were from line 357 to 361 as follows: “We mainly support a nutrient source hypothesis where the fungus can use the nutrient immobilized in leaf litter degrading also potential autotoxic compounds. The last is supported by the assumption that we observed increased root proliferation of ECM seedlings with conspecific litter. Oppositely if the fungal symbiont acts only as a protective layer surrounding root tips, it should not affect plant growth compared to ECM seedlings growing without litter. The presence of the basidiomycetous symbiont may prevent the…..”.
On the other hand, in our work, we do not specifically discuss allelopathy. Instead, we focus on autotoxicity, which is a component of the broader concept of allelopathy. Allelopathy is a term used to describe the effects of biochemicals released by plants on other plants, including themselves. However, in our study, we specifically examine autotoxicity, which refers to the effects of biochemicals produced by the same plant species, particularly those found in conspecific litter.
- After addressing the above issues, please rewrite the conclusion.
Done. Changes were included from line 384 to 386.
Reviewer 3 Report
Zotti et al. have investigated weather the ECM symbiosis can convert NF to PF due to plant litter. To reach their aim they assessed the effect of litter decomposition on seedling growth with and without mycorrhizal symbiosis. The manuscript is well written and the study point is interesting. I suggest acceptance for publication in Microorganisms Journal after minor revision to include:
Title: I suggest adding the impact of litter addition to plant-soil feedback.
Figure2: provide a figure with better resolution
Figure 4: Add the abbreviations to the figure legend
Author Response
Referee #3
Zotti et al. have investigated weather the ECM symbiosis can convert NF to PF due to plant litter. To reach their aim they assessed the effect of litter decomposition on seedling growth with and without mycorrhizal symbiosis. The manuscript is well written and the study point is interesting. I suggest acceptance for publication in Microorganisms Journal after minor revision to include:
We would like to thank the referee for their effort in evaluating our manuscript and for the positive recommendation.
Title: I suggest adding the impact of litter addition to plant-soil feedback.
Done. Following referee #2 suggestion as well, we changed title to “Quercus ilex ectomycorrhizal symbiosis with Pisolithus arrhizus shifts leaf-litter induced plant-soil feedback from negative to positive.”.
Figure2: provide a figure with better resolution
Done.
Figure 4: Add the abbreviations to the figure legend
Done.
Round 2
Reviewer 2 Report
I am satisfied with the authors' responses and agree to accept this manuscript.